# ZrTe$_2$/CrTe$_2$: an epitaxial van der Waals platform for spintronics

Yongxi Ou[1], Wilson Yanez [1], Run Xiao[1], Max Stanley[1], Supriya Ghosh [2], Boyang Zheng[1], Wei Jiang [3], Yu-Sheng Huang[1], Timothy Pillsbury[1], Anthony Richardella[1], Chaoxing Liu [1], Tony Low [3,4], Vincent H. Crespi [1], K. Andre Mkhoyan [2] & Nitin Samarth [1✉]

The rapid discovery of two-dimensional (2D) van der Waals (vdW) quantum materials has led to heterostructures that integrate diverse quantum functionalities such as topological phases, magnetism, and superconductivity. In this context, the epitaxial synthesis of vdW heterostructures with well-controlled interfaces is an attractive route towards wafer-scale platforms for systematically exploring fundamental properties and fashioning proof-of-concept devices. Here, we use molecular beam epitaxy to synthesize a vdW heterostructure that interfaces two material systems of contemporary interest: a 2D ferromagnet (1T-CrTe$_2$) and a topological semimetal (ZrTe$_2$). We find that one unit-cell (u.c.) thick 1T-CrTe$_2$ grown epitaxially on ZrTe$_2$ is a 2D ferromagnet with a clear anomalous Hall effect. In thicker samples (12 u.c. thick CrTe$_2$), the anomalous Hall effect has characteristics that may arise from real-space Berry curvature. Finally, in ultrathin CrTe$_2$ (3 u.c. thickness), we demonstrate current-driven magnetization switching in a full vdW topological semimetal/2D ferromagnet heterostructure device.

---

[1] Department of Physics and Materials Research Institute, The Pennsylvania State University, University Park, PA 16802, USA. [2] Department of Chemical Engineering and Materials Science, University of Minnesota, Minneapolis, MN 55455, USA. [3] Department of Electrical & Computer Engineering, University of Minnesota, Minneapolis, MN 55455, USA. [4] School of Physics & Astronomy, University of Minnesota, Minneapolis, MN 55455, USA. ✉email: nsamarth@psu.edu

Van der Waals (vdW) materials are an exciting playground for the discovery of emergent behavior in electrical, optical, and thermal properties in the two-dimensional (2D) limit and are potentially attractive for next-generation device applications[1–8]. The recent demonstration of long-range ferromagnetic order in 2D vdW materials has opened another new avenue to study magnetism in atomically thin films[9–15]. While many studies of 2D vdW ferromagnets have focused on mechanically exfoliated flakes[16–19], 2D vdW ferromagnets embedded in heterostructures create new opportunities for manipulating and engineering magnetic properties[20–22]. Such multilayer structures can potentially serve as building blocks for 2D magnetic and spintronics applications[14,15]. For example, chiral magnetic textures have been observed in mechanically stacked heterostructures using the vdW ferromagnet $Fe_3GeTe_2$[23,24]. Spin-orbit torque (SOT)-assisted magnetization switching has also been reported in layered vdW ferromagnets interfaced with heavy metals[25–27].

Amongst the candidate 2D ferromagnets, 1T-$CrTe_2$ has an interesting combination of properties. Bulk 1T-$CrTe_2$ is a known ferromagnetic material with a Curie temperature, $T_c$, above room temperature. This persists even in flakes exfoliated down to thicknesses of tens of nanometers[28–31]. Synthesized epitaxial thin films of this material also show a relatively high $T_c$ down to the quasi-2D regime[32,33]. An in-plane-to-out-of-plane transition of the magnetic easy axis in 1T-$CrTe_2$ may be controlled through thickness and strain in thin films[34,35]. Finally, 1T-$CrTe_2$ single crystals show reasonable stability against degradation after being exposed to the atmosphere[34].

Here, we report the synthesis by molecular beam epitaxy (MBE) of full vdW heterostructures that interface ultrathin ferromagnetic 1T-$CrTe_2$ films with $ZrTe_2$, a candidate topological Dirac semimetal. We note that a recent study[36] has demonstrated MBE growth of vdW heterostructures that epitaxially combine relatively thick (10 u.c.) $CrTe_2$ with a topological insulator ($Bi_2Te_3$) but we are not aware of any published reports of epitaxial vdW 2D ferromagnet/topological semimetal heterostructures. We use in vacuo angle-resolved photoemission spectroscopy (ARPES) to measure the band dispersion of metallic 1T-$CrTe_2$ and find that it is consistent with first-principles calculations. Measurements of the anomalous Hall effect (AHE) demonstrate robust ferromagnetism in both single 1T-$CrTe_2$ epilayers grown on sapphire and in vdW sapphire/$ZrTe_2$/$CrTe_2$ heterostructures. In thick films (12 u.c.) of 1T-$CrTe_2$ layers grown directly on sapphire or on $ZrTe_2$, we observe an AHE whose magnetic field dependence is suggestive of real-space Berry curvature effects. We further use the AHE to demonstrate the persistence of ferromagnetic order in one unit cell of 1T-$CrTe_2$ grown epitaxially on $ZrTe_2$, thus realizing a wafer-scale spintronics platform that epitaxially interfaces a 2D ferromagnet with a topological semimetal. Finally, we demonstrate current-induced magnetization switching in an ultrathin full vdW $ZrTe_2$/$CrTe_2$ heterostructure device, where the spin-orbit torque efficiency of the $ZrTe_2$ layer is evaluated via spin-torque ferromagnetic resonance (ST-FMR) measurements in $ZrTe_2$/$Ni_{80}Fe_{20}$ (permalloy, Py) heterostructures.

## Results and discussion
### MBE growth and characterizations of the $ZrTe_2$/$CrTe_2$ heterostructures.
We grew single-layer 1T-$CrTe_2$ and $ZrTe_2$/$CrTe_2$ (Fig. 1a) heterostructures on (001) sapphire substrates by co-deposition from Cr (Zr) and Te sources in an MBE chamber with a base pressure of $\sim 1 \times 10^{-10}$ mbar. The growth was monitored with 13 keV reflection high energy electron diffraction (RHEED). Sharp streaky RHEED patterns (Fig. 1b) indicated the epitaxial

growth of the materials (see Materials and Methods for details). The film thickness during growth was controlled by the deposition time (~1 u.c. per 10 min for $CrTe_2$ growth, ~1 u.c. per 25 min for $ZrTe_2$) as calibrated from x-ray reflectometry. To protect the thin films from oxidation during ex situ characterization, we deposited a capping layer of ~40 nm Te. The 1T-$CrTe_2$ crystal structure belongs to the $P\bar{3}m1$ space group (Fig. 1a). We characterized the crystalline structure of the $ZrTe_2$/$CrTe_2$ heterostructures using aberration-corrected scanning transmission electron microscopy (STEM), as shown in the cross-sectional high-angle annular dark-field (HAADF) images in Fig. 1c, revealing an atomically flat interface between $ZrTe_2$ and $CrTe_2$. The atomic alignment between the $CrTe_2$ layers matches well to that of 1 T phase $CrTe_2$ (Fig.1c). We used energy dispersive X-ray (EDX) spectroscopy to determine the relative concentration of Cr and Te in the $CrTe_2$ thin films, where it showed Cr/Te=0.53, indicating limited (if any) Cr intercalation (See Supplementary information). The lattice constants of the 1T-$CrTe_2$ thin films grown on sapphire, $a = b = 3.93 \pm 0.06$ Å and $c = 6.02 \pm 0.05$ Å ($\langle 90^o \times 90^o \times 120^o \rangle$), were further evaluated from the atomic HAADF-STEM images using sapphire as a reference. The measured in-plane lattice constant is ~3% larger than reported in bulk crystals[28], possibly because our epitaxial layers are strained. Figure 1d shows the topography of 1 u.c. $CrTe_2$ grown on $ZrTe_2$ measured via in vacuo transfer to a scanning tunneling microscope (STM). The height profile indicates the thickness of ~6.0 Å for the 1 u.c. $CrTe_2$ sample, in good agreement with the TEM results.

Figure 1e shows the x-ray diffraction spectrum of a $CrTe_2$ thin film with peaks corresponding to the out-of-plane (001) growth direction as well as peaks from the sapphire substrate. The rocking curve (inset of Fig. 1e) of the $CrTe_2$ (001) peak gives a full width at half maximum (FWHM) ~0.03 degree, indicating a reasonable crystallinity in the grown films. Additional reciprocal lattice maps were used to characterize the mosaic spread (see Supplementary information). We also used X-ray photoemission spectroscopy (XPS) to determine the sample composition. Figure 1f shows interference between Te 3$d$ and Cr 2$p$ XPS spectrum. The presence of chromium was also confirmed by the weak Cr 3$s$ and Cr 2$s$ peaks (not shown here) as well as curve fitting of reference telluride and Cr° spectra acquired under similar conditions. Peak positions obtained were as follows: Te 3$d_{5/2}$ at 572.0 eV, Te 3$d_{3/2}$ at 582.4 eV, Cr 2$p_{3/2}$ at 573.5 eV, and Cr 2$p_{1/2}$ at 582.7 eV respectively. The XPS element analysis gives the Te/Cr concentration ratio ~2:1, which is in good agreement with the STEM-EDX analysis.

We measured the band structure of the 1T-$CrTe_2$ thin films through in vacuo transfer to an ARPES chamber with excitation from the 21.2 eV I$\alpha$ spectral line of a helium plasma lamp. Photoemitted electrons were detected by a Scienta Omicron DA 30 L analyzer with 6 meV energy resolution. Figure 2a shows the hexagonal Brillouin zone of 1T-$CrTe_2$ and Fig. 2b shows the band dispersion of the 1T-$CrTe_2$ (001) surface measured along the $\bar{\Gamma} - \bar{M}$ direction at room temperature. The location of the chemical potential within the valence bands indicates the sample is p-type, a fact also confirmed using Hall effect measurements. We performed first-principles density functional theory (DFT) calculations for bulk 1T-$CrTe_2$ including spin-orbit coupling with the magnetic moment oriented out-of-plane, obtaining the band structure shown in Fig. 2c. There is good agreement between the ARPES spectrum and the calculated bands (see Supplementary information for more 1T-$CrTe_2$ calculations). The asymmetry observed in the ARPES intensity is likely due to matrix element effects[37].

Figure 2d, e show the ARPES spectra of certain $ZrTe_2$/$CrTe_2$ heterostructures. The ARPES spectrum of ultrathin 1 u.c. $CrTe_2$

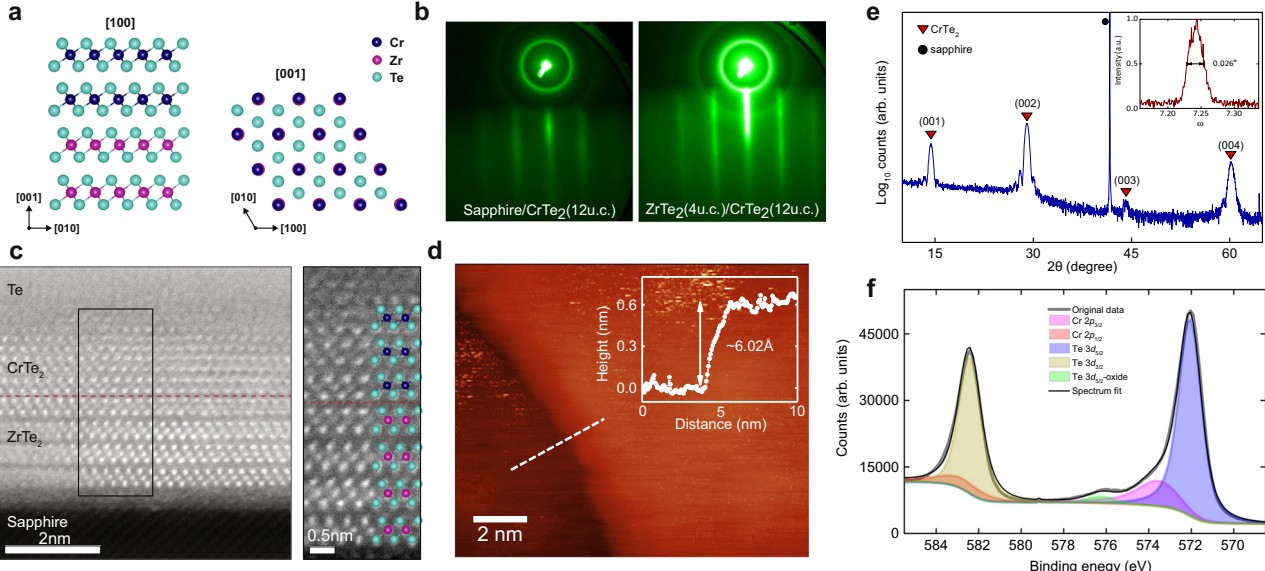

**Fig. 1 MBE and characterization of 1T-CrTe₂ thin films. a** Schematics of the 1 T phase of CrTe₂ grown on ZrTe₂. Note that [100], [010], and [001] correspond to $[1\bar{2}10]$, $[2\bar{1}\bar{1}0]$, and [0001] in the 4 axis Miller–Bravais notation. **b** RHEED patterns of the single layer of CrTe₂ (12 u.c.) and ZrTe₂ (4 u.c.)/ CrTe₂ (12 u.c.) heterostructure. The substrate is sapphire in both cases and the electron beam is directed along the [11$\bar{2}$0] orientation of sapphire. **c** HAADF-STEM images of the ZrTe₂ (4 u.c.)/CrTe₂ (3 u.c.) heterostructure viewed in cross-section. The images have been low-pass filtered for clarity. **d** STM image of a ZrTe₂ (4 u.c.)/CrTe₂ (1 u.c.) sample. The line scan shows the thickness of the 1 u.c. CrTe₂. **e** XRD 2θ scan of a 12 u.c. CrTe₂ thin film. **f** XPS spectrum of an 18 u.c. thick CrTe₂ film. The small Te oxide shoulder is due to the absence of a capping layer in this sample. The counts in **e** and **f** are in arbitrary units (arb.units.).

grown on ZrTe₂ largely resembles the band dispersion of ZrTe₂. We attribute this to the CrTe₂ 1 u.c. layer (~0.6 nm) being thinner than the mean free path of photoemission electrons near the sample surface, rendering the ARPES signal from the ZrTe₂ layer beneath still detectable. Note the linear dispersion of the Dirac band from ZrTe₂, supporting the presence of a topological Dirac semimetal phase of the 4 u.c. ZrTe₂ film[38,39]. As the CrTe₂ film becomes thicker (3 u.c. CrTe₂), its ARPES band dispersion looks more similar to the 12 u.c. CrTe₂ results. The smooth transition of the band dispersion in the ZrTe₂/CrTe₂ heterostructure suggests an excellent epitaxy between the two materials and lays the foundation for the observed robust ferromagnetic order in such bilayers as we discuss below.

**Anomalous Hall resistance measurements.** Next, we describe transport measurements of 1T-CrTe₂ epilayers and ZrTe₂/CrTe₂ heterostructures. We measured the Hall resistance of the samples as a function of an out-of-plane magnetic field at various temperatures (details of the longitudinal resistivity measurements are given in the Supplementary information). Figure 3 shows the sample schematics as well as the Hall resistance of the single layer and heterostructures (see Methods for measuring and analyzing the Hall resistance data). Starting with the results in a 12 u.c. (~7.2 nm thick) CrTe₂ film grown on sapphire, the Hall effect at room temperature shows a small nonlinear Hall signal at low magnetic fields, suggesting weak ferromagnetic order with a Curie temperature ($T_c$) in the vicinity of room temperature. This is consistent with an earlier report that a 10 nm thick exfoliated flake of 1T-CrTe₂ has a $T_c$ above room temperature[29]. Note that earlier reports indicate that $T_c$ may be enhanced in thin films of CrTe₂ and other CrTe compounds in CVD-grown samples as compared to thicker films[34,40].

As we cool the 12 u.c. CrTe₂ thin film below 200 K, a stronger AH resistance appears with a hysteresis loop whose coercivity continually increases as the temperature falls below 100 K. The hysteretic AHE loop indicates that the 1T-CrTe₂ thin film

exhibits an out-of-plane magnetic easy axis. As we discuss later, we observed an out-of-plane magnetic easy axis in all our CrTe₂ thin films down to the 1 u.c. limit. This contrasts with the in-plane easy axis observed in bulk exfoliated CrTe₂ flakes[29–31] but agrees with other thin-film results[32,34,36]. To understand the magnetic anisotropy behavior in our samples, we used DFT to compute the magnetic anisotropy energy of 1T-CrTe₂ thin films with different lattice constants (see Supplementary information). We find that the easy axis is sensitive to both lattice strain, in agreement with a previous prediction[35], and Fermi-level position, making it possible to have different easy axis directions in different experimental settings.

As the sample temperature further decreases below around 40 K, the hysteresis loop of the AHE is replaced by an unconventional shape that has a non-monotonic dependence on the magnetic field, vanishing at both low and high fields and with a peak at an intermediate field. This feature becomes more prominent at low temperatures (2 K). This unusual feature in the Hall resistance has been reported in other ferromagnetic systems where the ferromagnetic order can be of intrinsic origin[41–43] from ferromagnetic doping[44,45] or from an interfacial proximity effect[46]. It is regarded as a sign of the topological Hall effect (THE), arising from complex real-space chiral domain structures such as skyrmions[47,48]. This unconventional Hall effect is also seen in CrTe₂ films of the same thickness grown on ZrTe₂ (Fig. 3c). We note that a recent study of the AHE in CrTe₂/Bi₂Te₃ has also shown similar magnetic field dependence and has been interpreted in terms of a THE arising from the presence of a non-collinear inversion-symmetry-breaking Dzyaloshinskii–Moriya (DM) interaction due to the interplay between the strong spin-orbit coupling at the CrTe₂/Bi₂Te₃ interface[36]. This interpretation was supported by theoretical simulations. Our observation of a THE-like signature in CrTe₂ films grown directly on sapphire (a material that does not have strong spin-orbit coupling) suggests that the underlying physics is probably more complex. We might speculate that the interface between CrTe₂ and

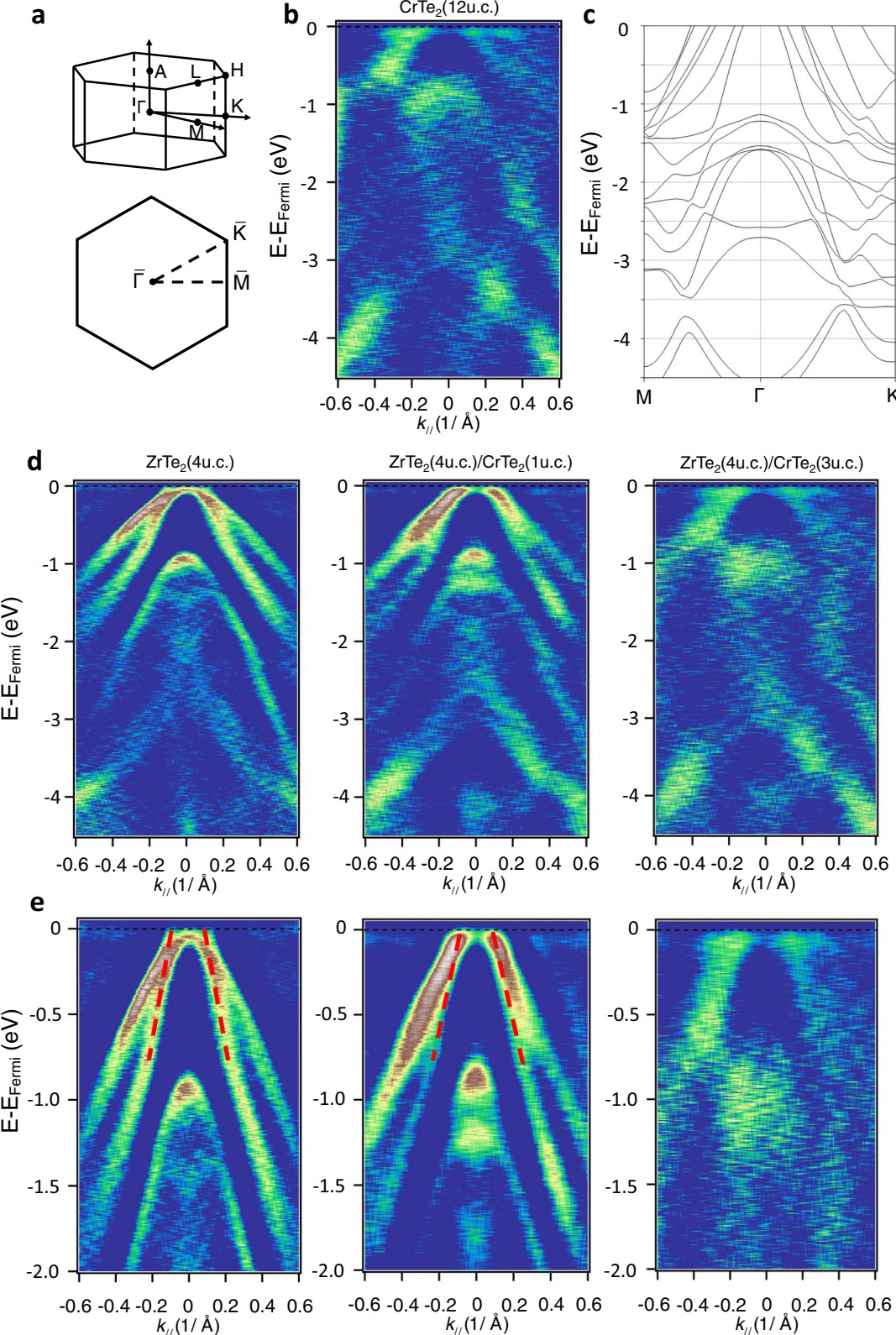

**Fig. 2 ARPES measurements and DFT calculation of the band structure of 1T-CrTe₂ thin films. a** Schematic of the bulk and projected Brillouin zones of CrTe₂. **b** ARPES spectrum of a single layer of 12 u.c. CrTe₂ in the $\bar{\Gamma} - \bar{M}$ direction. **c** DFT calculation of the band structure of bulk CrTe₂. The M and K points are 0.92 and 1.06 Å⁻¹ respectively. **d**, **e** ARPES spectrum of the 4 u.c. ZrTe₂ (left), ZrTe₂ (4 u.c.)/CrTe₂ (1 u.c.) (middle), and ZrTe₂ (4 u.c.)/CrTe₂ (3 u.c.) (right) with a larger (**d**) and smaller (**e**) binding energy scale. All the ARPES data were taken at 300 K with 21.2 eV excitation from a He lamp. To more clearly highlight the measured band dispersion, we present all plots as second-derivatives with respect to the energy. The red dashed lines are guides to the eyes for the Dirac dispersion in ZrTe₂.

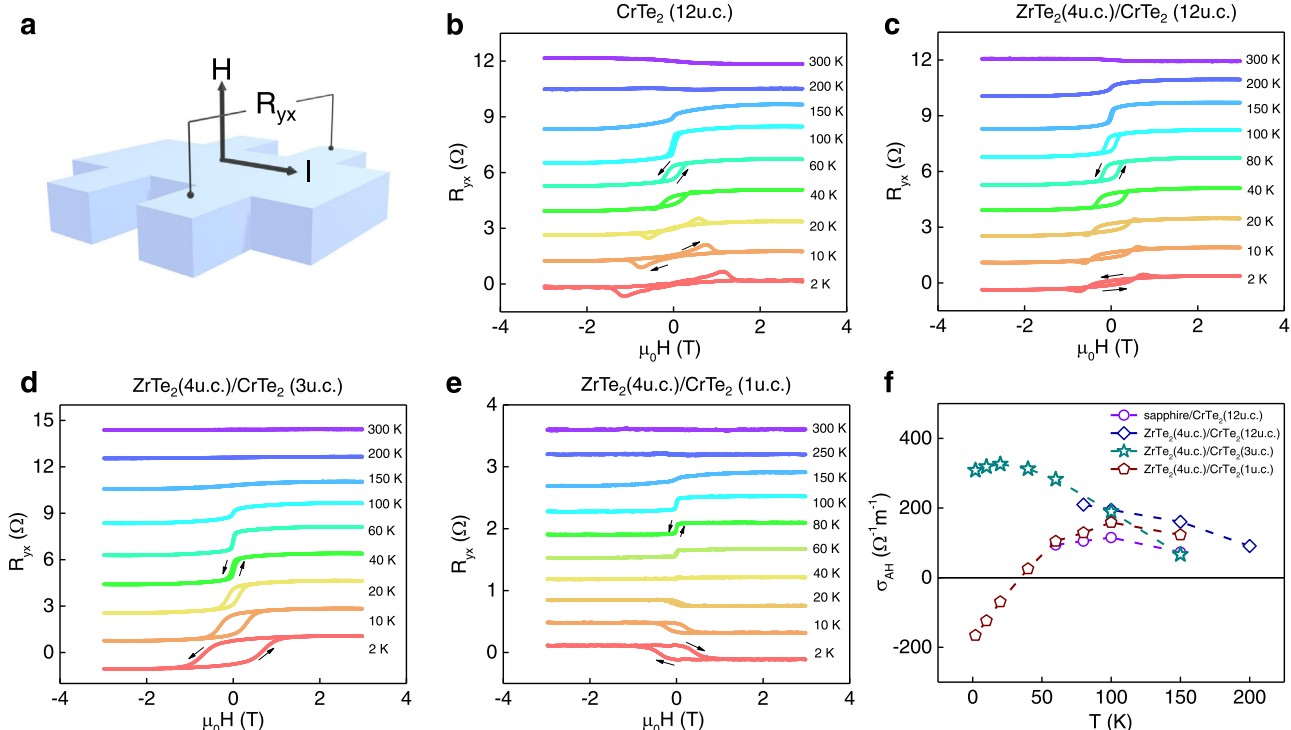

**Fig. 3 Anomalous Hall resistance measurements on CrTe$_2$ layers and ZrTe$_2$/CrTe$_2$ heterostructure. a** Schematics of the Hall bar. **b–e** Anomalous Hall resistance (AHR) of the 12 u.c. CrTe$_2$ (**b**), ZrTe$_2$ (4 u.c.)/CrTe$_2$ (12 u.c.) (**c**), ZrTe$_2$ (4 u.c.)/CrTe$_2$ (3 u.c.) (**d**), and ZrTe$_2$ (4 u.c.)/CrTe$_2$ (1 u.c.) (**e**). The AH resistance data has been offset for clarity. The arrows denote the field sweep directions. **f** Anomalous Hall conductivity of the samples shown from **b** to **e**.

sapphire induces a DM interaction but we do not currently have any microscopic model to justify this speculation. The magnitude of the THE in our CrTe$_2$/sapphire films is a factor of 6 smaller than that reported in CrTe$_2$/Bi$_2$Te$_3$. Notably, exfoliated CrTe$_2$ flakes have been shown to exhibit Néel-type domain walls due to the sixfold crystalline symmetry[31]. Since our CrTe$_2$ thin films exhibit an out-of-plane magnetic easy axis, we posit that these Néel-type domain walls transform into a chiral magnetic texture that is an inherent property of CrTe$_2$ itself, producing the observed behavior in the Hall measurement. At this stage, we cannot definitively rule out alternative scenarios such as competing ferromagnetic phases that produce AHE of opposite sign[49,50]. However, a detailed analysis of the variation of the Hall effect as a function of temperature and magnetic field suggests that such a trivial alternative scenario is unlikely (see Supplementary information for more discussion). Direct real-space experimental evidence of the chiral domain structures, such as low-temperature magnetic force microscopy or Lorentz transmission electron microscopy, will be needed to definitively determine the nature of the magnetic ordering and its impact on the Hall effect.

We now discuss the ferromagnetism in thinner CrTe$_2$ films, focusing on sapphire/ZrTe$_2$/CrTe$_2$/Te heterostructures that are of higher structural quality than ultrathin CrTe$_2$ films grown directly on sapphire. Our measurements show that 3 u.c. thick CrTe$_2$ films only begin to show an AHE below 150 K (Fig. 3d). Measurements down to 2 K show a conventional AHE whose magnitude increases monotonically with decreasing temperature; we do not observe any unconventional signatures in the AHE unlike in 12 u.c. thick CrTe$_2$ films. We note that a thickness-dependent THE has been observed in other material systems[42,43], including another CrTe compound epitaxially grown on SrTiO$_3$(111) substrates[51]. In our CrTe$_2$ films, we tentatively attribute the observed thickness dependence of the Hall effect to the enhancement in magnetic anisotropy which prefers an out-of-

plane easy axis that arises from a weakening of the Coulomb screening effect in the 2D limit[34]. We speculate that the 3 u.c. CrTe$_2$ film in this case may have different anisotropy energy compared to the thicker sample, such that it no longer satisfies the anisotropy requirement to form a proper phase to exhibit chiral magnetic texture.

To test the robustness of the ferromagnetic order in the true 2D limit of 1T-CrTe$_2$, we also measured a heterostructure consisting of only 1 u.c. of CrTe$_2$ grown on ZrTe$_2$ (Fig. 3e). Like the 3 u.c. CrTe$_2$ film, an AHE signal appears at around 150 K. The observation of the AHE unambiguously demonstrates the existence of long-range ferromagnetism even down to the 1 u.c. limit of CrTe$_2$ in the heterostructure, confirming vdW 1T-CrTe$_2$ as a 2D ferromagnet. In contrast to the thicker films, the magnitude of the AHE shows a non-monotonic dependence on temperature. As the temperature is lowered from 150 K, the magnitude initially increases, reaching a maximum at around 100 K. It then decreases with further lowering of temperature, vanishing at around 40 K and then increasing again but with the opposite sign. This sign reversal of the AHE may be a consequence of the variation of the Berry curvature[52] induced by the charge transfer between ZrTe$_2$ and CrTe$_2$. A similar phenomenon has been reported previously in magnetic topological insulator heterostructures[53]. Figure 3f summarizes the measured temperature-dependent anomalous Hall conductivity of the various CrTe$_2$ samples (see Supplementary information for more results on the longitudinal magnetoresistance and magnetometry measurements). The difference in the strength of the AHE between sapphire/CrTe$_2$ and ZrTe$_2$/CrTe$_2$ samples might be due to the various interfacial contributions to the AHE and THE, an effect that has been known in other ferromagnetic heterostructures[54]. Without a more detailed microscopic model of the interfacial band structure and its Berry curvature contributions to spin transport, it is not possible for us to

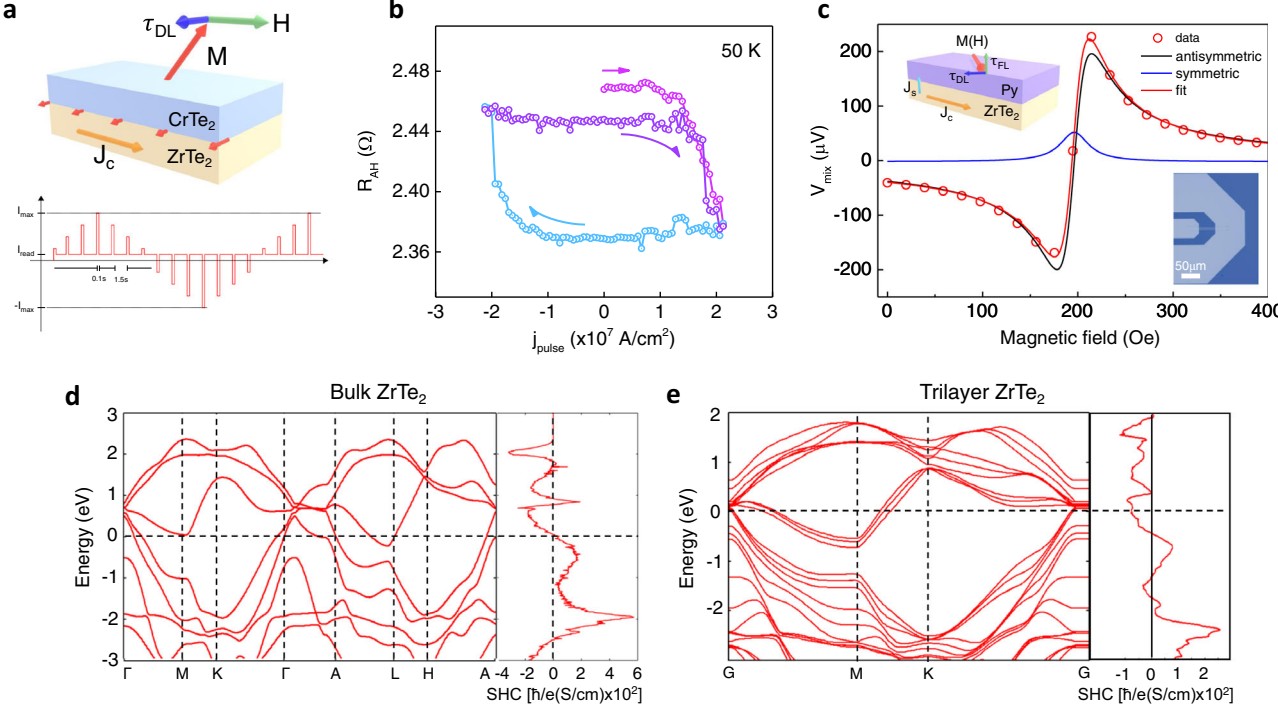

**Fig. 4 Pulsed current-induced magnetization switching of an ultrathin ZrTe₂/CrTe₂ heterostructure device and SOT characterizations of ZrTe₂.**
**a** Schematics of the SOT assisted magnetization switching in the ZrTe₂/CrTe₂ heterostructure and the writing and reading pulse current sequence. **b** Pulse current-induced magnetization switching of a ZrTe₂ (8 u.c.)/CrTe₂ (3 u.c.) at 50 K under an external magnetic field (700 Oe) applied in-plane parallel to the current direction. **c** ST-FMR spectrum of a ZrTe₂/Py bilayer heterostructure at room temperature. Inset: Optical microscope image and schematic of the ST-FMR device. **d**, **e** DFT calculated band structure and spin Hall conductivity of ZrTe₂ in the bulk and thin film form, respectively.

provide a more definitive explanation of the variation of the AHE and THE in these different interfacial configurations. We also caution that current shunting through the ZrTe₂ layer adds complications to a direct comparison of the magnitude of the THE in these different heterostructures.

**Current-induced spin-torque and magnetization switching.** Finally, we examine a proof-of-concept spintronic device demonstration of our wafer-scale vdW Dirac semimetal/2D ferromagnet. By using optical lithography, we fabricated 5 μm × 10 μm scale Hall bar devices of a ZrTe₂(8 u.c.)/CrTe₂(3 u.c.) heterostructure and used the AHE to probe the response of the CrTe₂ magnetization to the current flowing in the heterostructure (Fig. 4a). We note that the ZrTe₂ and CrTe₂ films have similar conductivities so that current flows in parallel in the two layers. Figure 4b show magnetization switching at 50 K under a pulsed longitudinal current with an external field 700 Oe parallel to the current direction. An out-of-plane magnetic field was first applied to align the magnetization of CrTe₂ to an initial state before using a pulsed current to switch the magnetization (see Methods). As shown in Fig. 4b, positive and negative current pulses switch the magnetization between two states; the switching edge appears to be step-like, indicating the switching likely involves multi-domain nucleation and expansion under the current-induced spin-orbit torque (SOT) from ZrTe₂. The average current density for the onset of the magnetization switching of this ZrTe₂(8 u.c.)/CrTe₂(3 u.c.) device is about $1.8 \times 10^7$ A/cm², which is comparable to the current density needed to switch the magnetization of 3D ferromagnets (e.g., Co, CoFeB) using heavy metals[55,56]. However, we emphasize that because of the complicated nature of the domain nucleation and domain wall motion during the magnetization switching process in micron-meter size devices, it is difficult to directly evaluate the SOT from the switching current

density alone[27,56]. We note that we have observed current-switching SOT switching over the temperature range 10 K < T < 90 K without much variation in the threshold switching current density (see Supplementary information for results at additional temperatures and bias fields). Technical constraints prevent us from directly measuring the SOT efficiency using techniques such as ST-FMR and spin pumping at the low temperature required by the Curie temperature of the 3 u.c. CrTe₂. For a better understanding of the efficiency of ZrTe₂ as a SOT material, we instead carried out ST-FMR measurements at room temperature on a ZrTe₂/permalloy (Py) heterostructure. Figure 4c shows the ST-FMR data measured at room temperature on a 50 μm × 10 μm ZrTe₂(5 nm)/Py(4 nm) device (see Methods). This mixing voltage ($V_{mix}$) signal is the result of the dynamics of the magnetization of the Py layer driven by the SOT induced in the ZrTe₂ layer by the input radio-frequency (RF) current. The resonance shape of $V_{mix}$ can be separated into a symmetric (S) and antisymmetric (A) Lorentzian component respectively, where the symmetric (antisymmetric) component is proportional to the current-induced SOT (Oersted field). The SOT efficiency, $\xi_{FMR}$, defined as the ratio of the spin current ($J_s$) to the charge current ($J_c$), is evaluated from the ratio of the symmetric and antisymmetric components[57]: $\xi_{FMR} = \frac{2e}{\hbar}\frac{J_s}{J_c} = \frac{S}{A}\frac{e\mu_0 M_s t_{ZrTe_2} t_{Py}}{\hbar}\left[1 + \left(\frac{M_{eff}}{H_{Res}}\right)\right]^{1/2}$, where $e$ is the charge of the electron, $\hbar$ is the reduced Planck constant, $\mu_0$ is the permeability of free space, $M_S$ is the saturation magnetization of Py, $t_{ZrTe2(Py)}$ is the thickness of the ZrTe₂(Py) layer, $M_{eff}$ is the effective magnetization and $H_{Res}$ is the resonance field respectively. From the data in Fig. 4c, we obtain $\xi_{FMR} = 0.014 \pm 0.005$ for ZrTe₂/Py. With the measured electrical conductivity of ZrTe₂, $\sigma_{xx}^{ZrTe_2} = 3.16 \times 10^5 \text{Sm}^{-1}$, the effective spin Hall conductivity (SHC) of such a ZrTe₂ thin film is estimated as: $\sigma_{SH,effective}^{ZrTe_2} = (\hbar/2e)\sigma_{xx}^{ZrTe_2}\xi_{FMR} \approx (\hbar/e)2.2 \times 10^3 \text{Sm}^{-1}$. While

the ST-FMR result clearly demonstrates charge-to-spin conversion in the Dirac semimetal $ZrTe_2$ layer and shows that the spin current generated in the $ZrTe_2$ layer is playing an important role in the current-induced magnetization switching experiment in $ZrTe_2/CrTe_2$ heterostructures, the SOT efficiency deduced in this manner is usually only a lower bound of the full SOT efficiency generated inside the SOT material due to the non-ideal interface and interfacial spin transparency. Since the interface in $ZrTe_2/CrTe_2$ is a more coherent one compared to that in $ZrTe_2/Py$, we expect a more efficient spin current transfer in the former because of the epitaxial interface and the smooth transition of the band structure as indicated by our ARPES measurements (Fig. 2).

To obtain further insights into the spin-charge conversion generated by $ZrTe_2$, we also carried out first-principles calculations of the SHC of both bulk and multilayer $ZrTe_2$ (Fig. 4d, e). For the bulk phase, the SHC near the Fermi level is almost zero, and increases in magnitude with electron or hole doping, which is consistent with our relatively small effective SHC of $ZrTe_2$ determined via ST-FMR (see above). On the other hand, for the trilayer case, the Fermi level shifts to higher energy with a noticeable broad SHC peak (Fig. 4e). The origin of this SHC peak can be attributed to the presence of topological Dirac nodes in the vicinity of the Fermi level. We have also calculated the SHC for monolayer and bilayer $ZrTe_2$. The bilayer shows similar behavior as a trilayer while the monolayer shows quite different behavior (See Supplementary information). Note that the actual Fermi level of $ZrTe_2$ may be influenced by the presence of the Py layer, due to charge transfer that results from a difference in their work functions. Nevertheless, due to the broad SHC peak in trilayer $ZrTe_2$, we anticipate the Fermi level to be within this conductivity peak.

In conclusion, we used MBE to synthesize 1T-$CrTe_2$ thin films and full vdW wafer-scale 1T-$CrTe_2/ZrTe_2$ heterostructures. The out-of-plane magnetic easy axis of these MBE-grown ferromagnetic $CrTe_2$ films is of particular interest for studying proof-of-concept spintronics applications such as perpendicular magnetic tunnel junctions and will be technologically relevant once a robust room temperature ferromagnetic state is realized. We observed behavior consistent with a THE in both $CrTe_2$ epilayers and heterostructures as indicated by AHE measurements, suggesting $CrTe_2$ as a promising material platform for studies of chiral magnetic domain structures. We also demonstrated that long-range ferromagnetic order persists in the heterostructure $ZrTe_2/CrTe_2$ down to the 1 u.c. limit of 1T-$CrTe_2$, and we further demonstrated current-induced magnetization switching in an ultrathin $ZrTe_2/CrTe_2$ full vdW heterostructure and characterized the SOT from $ZrTe_2$ via ST-FMR measurements. The wafer-scale epitaxial synthesis of heterostructures that cleanly interface the vdW 2D ferromagnet $CrTe_2$ with the topological semimetal $ZrTe_2$ may provide new opportunities in studying the coexistence of the 2D ferromagnetic and topological phases, interfacial interactions such as proximity induced magnetism in vdW topological semimetals, as well as the SOC interaction and spin-torque phenomena in the true 2D limit.

## Methods

**Sample growth**. We deposited single layer 1T-$CrTe_2$ thin films and $ZrTe_2$/1T-$CrTe_2$ thin film heterostructures using MBE in a Scienta Omicron EVO50 system under ultrahigh vacuum (~$10^{-10}$ mB). The sapphire (0001) substrates were outgassed at 600 °C in situ for 1 h to clean the surface before the deposition of the thin films. The epitaxial 1T-$CrTe_2$ was grown at a substrate temperature of 280 °C via co-evaporation of Te (purity: 99.99%, Alfa Aesar) and Cr (purity: 99.997%, Alfa Aesar) respectively, with the flux ratio ~40:1. $ZrTe_2$ was grown at a substrate temperature of 420 °C. Te was sublimated at a significant overpressure compared to the Zr (purity: grade 702, Kurt J.Lesker) which was evaporated via e-beam at a deposition rate of roughly 0.3Å/min. During the growth of $ZrTe_2$, the film was annealed periodically throughout the growth under a constant tellurium flux in order to avoid vacancies and mitigate defects. The outgassing and growth

temperatures were measured by an infrared camera with an emissivity of 0.7. RHEED was monitored using a 13 keV electron gun during the growth of the samples. Before ex situ characterization and measurements, we capped the samples with 40 nm Te to avoid degradation.

**STM and ARPES measurements**. In situ topography was measured at 300 K after transferring MBE-grown samples in vacuo to a Scienta Omicron LT NANOPROBE STM system. We also carried out ARPES measurements at 300 K after in vacuo transfer following the MBE growth of the samples. As excitation, we used the 21.2 eV spectral line from a helium plasma lamp and the emitted photoemission electrons were detected by a Scienta Omicron DA 30 L analyzer with an energy resolution of 6 meV.

**STEM characterization and analysis**. Cross-section samples for the STEM study were prepared on an FEI Helios Nanolab G4 dual-beam Focused Ion Beam (FIB) system with 30 keV Ga ions followed by ion-milling at 2 keV to removed damaged surface layers. Amorphous C and Pt were first deposited on the films to protect the surface from damage on exposure to the ion beam. STEM experiments were performed on an aberration-corrected FEI Titan G2 60–300 (S)TEM microscope, which is equipped with a CEOS DCOR probe corrector, monochromator, and a super-X energy dispersive X-ray (EDX) spectrometer. The microscope was operated at 200 and 300 keV with a probe current of 80 pA. HAADF-STEM images were acquired with the probe convergence angle of 25.5 mrad and the detector inner and outer collection angles of 55 and 200 mrad respectively. EDX elemental maps were acquired and analyzed using Bruker Esprit software. The lattice constants were obtained by using the Fourier transform atomic-resolution HAADF-STEM images.

**XRD and XPS characterization**. We carried out XRD measurements on X'Pert³ MRD operating in the reflection mode with Cu-Kα radiation (45 kV, 40 mA) and diffracted beam monochromator, using a step scan mode with the step of 0.025°(2θ) and 0.88 s per step. The XPS experiments were performed using a Physical Electronics VersaProbe II instrument equipped with a monochromatic Al kα x-ray source (hν = 1,486.6 eV) and a concentric hemispherical analyzer. Peaks were charge referenced to the $CH_x$ band in the carbon 1 s spectra at 284.8 eV. Measurements were made at a takeoff angle of 45° with respect to the sample surface plane. A model line shape for the Te 3d spectrum was determined from an exfoliated, oxygen-free $Bi_2Te_3$ sample[58]. We assumed that the shapes would be very similar. The Cr 2p line shape was modeled using reference Cr° spectra from the instrument vendor. Three sets of highly constrained doublets (1 each for Cr, Te°, and $TeO_x$) were used for the Cr 2p-Te 3d region. Quantification was done using instrumental relative sensitivity factors (RSFs) that account for the X-ray cross-section and inelastic mean free path of the electrons. The analysis region was ~200 μm in diameter. The sapphire/$CrTe_2$ sample without capping layers as measured by XPS was transferred after removal from the MBE chamber into the XPS instrument within 5 min to minimize oxidation.

**Electrical transport characterization**. We performed electrical transport measurements in a Quantum Design physical properties measurement system (PPMS) in a Hall bar configuration. Hall bars with lateral dimensions of 1 mm × 0.5 mm were mechanically defined. The Hall resistance of the devices, $R_{yx}^*$, was measured as a function of magnetic field up to 3 T in the temperature range between 2 and 300 K. To determine the anomalous Hall response at a given temperature, we first antisymmetrized the magnetic field dependence of the Hall resistance to remove the longitudinal resistance contribution; then, we subtracted the ordinary Hall resistance $R_{H_0}$: $R_{yx} = \frac{\left(R_{yx}^*(H,T) - R_{yx}^*(-H,T)\right)}{2} - R_{H_0}$. For samples that do not exhibit any THE, the AH resistance $R_{AHE}$ is then equal to $R_{yx}$. For samples that show a THE signal, the total transverse resistance can be written as $R_{yx} = R_{AHE} + R_{THE}$. The anomalous Hall conductivities $\sigma_{AH}$ are calculated as $\sigma_{AH} = \frac{\rho_{AHE}}{\left(\rho_{AHE}^2 + \rho_{xx}^2(H=0)\right)}$.

**Pulsed switching measurement**. We carried out pulsed current-induced magnetization switching experiments in a PPMS using an external Keithley 2450 source meter and a Keithley 2182 A nanovoltmeter. Before each switching attempt, the magnetization of the device was set and saturated in an initial state by applying a perpendicular magnetic field. In the pulse switching measurement, a train of pulses consisting of a 100 ms current pulse of varying magnitude followed by a 1500 ms pulse of 100 μA was applied under a magnetic field parallel to the current direction, during which we measured the anomalous Hall resistance of our system.

**Spin-torque ferromagnetic resonance measurement**. To further study the charge-to-spin conversion in the Dirac semimetal $ZrTe_2$ layer, we have performed ST-FMR in a $ZrTe_2$/permalloy heterostructure. Without breaking the vacuum, we synthesized $ZrTe_2$ (5 nm)/Py (4 nm)/Al (4 nm) heterostructures. These heterostructures were then patterned into 50 um × 10 um bars using standard lithography techniques including a two-step plasma etching process using $BCl_3$ and Ar as precursor gases. ST-FMR measurements were performed in these devices using a probe station equipped with a GMW 5201 projected field electromagnet, a Keysight E8257D analog signal generator and a Keithley 2182 A nanovoltmeter. The

spectrum was measured using a radiofrequency current ranging from 4 GHz to 6 GHz with an applied in-plane magnetic field up to 1.6 KOe.

**DFT first-principles calculation.** Spin-orbit-coupled (SOC) DFT calculations were implemented in the Vienna Ab-initio Simulation Package (VASP)[59–61] and Quantum Espresso[62]. The lattice constant for 1T-phase $CrTe_2$ and $ZrTe_2$ was obtained from the experimental result. The z-axis cell dimension is 15 Å for monolayer $CrTe_2$ to isolate a layer from its periodic images. The exchange-correlation is treated under GGA PBE approximation[63] with PAW method[64]. The energy cutoff in all calculations was 500 eV, and the k-point sampling was set as $16 \times 16 \times 1$ and $16 \times 16 \times 10$ centered at Γ for monolayer and bulk structures. The residual force after relaxation is smaller than 0.01 eV/A for all atoms. DFT+U method[65,66] is used in the calculation and the effective U is 2 eV to make results comparable to previous works[67,68]. Spin Hall conductivity is calculated based on kubo formula using the fitted Hamiltonian, as implemented in the Wannier90 package[69].

## Data availability

All data for the figures and other Supplementary information that support this work are available upon reasonable request to the corresponding author.

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

## Acknowledgements

The MBE synthesis, ARPES, STM measurements, and theoretical calculations were supported by the Penn State Two-Dimensional Crystal Consortium-Materials Innovation Platform (2DCC-MIP) under NSF Grant No. DMR-2039351 (Y.O., B.Z., M.S., T.P., A.R., V.H.C., and N.S.). Transport measurements were carried out under the support of the Institute for Quantum Matter under DOE EFRC grant DE-SC0019331 (R.X. and N.S.). The TEM, XRD, SOT measurements, and spin Hall conductivity calculations were supported by SMART, one of seven centers of nCORE, a Semiconductor Research Corporation program, sponsored by the National Institute of Standards and Technology (NIST) (W.Y., Y.-S.H., S.G., T.L., K.A.M., and N.S.). Parts of this work were carried out in the Characterization Facility, the University of Minnesota, which receives partial support from the NSF through the MRSEC (Award Number DMR-2011401) and the NNCI (Award Number ECCS-2025124) programs (S.G.and K.A.M.) We thank Jeffrey Shallenberger for assistance in XPS measurements and Hemian Yi for helpful discussions about ARPES measurements.

## Author contributions

Y.O. and N.S. conceived the project and experiments. Y.O. and M.S. grew the samples and perform the ARPES measurements. R.X. and M.S. performed the transport measurements. W.Y. performed the pulsed current-induced magnetization switching with assistance from Y.O. M.S., Y.-S.H., T.P., and A.R. characterized the samples. S.G. conducted and analyzed the TEM data under the supervision of K.A.M. B.Z. performed the DFT band structure calculation under the supervision of V.H.C. W.J. and T.L. performed the spin Hall conductivity calculations. Y.O. and N.S. wrote the manuscript with substantial contributions from all authors.

## Competing interests

The authors declare no competing interests.
