## [Peer Review File · Nature Communications]

Reviewers' Comments:

Reviewer #1:

Remarks to the Author:

The authors demonstrate an interesting epitaxy of heterojunction consisting of topological and 2d ferromagnetic materials. The the growth quality is high and the details are sufficient for repeating the results. The author also performed systematic hall measurement on the samples of pure CrTe₂ and CrTe₂/ZrTe₂ interfaces, revealing the anomalous Hall effect together with topological Hall effects. Before accept the author's explanation, I want the author display the magnetization results together with longitude magnetic resistance measurements for the above samples at corresponding temperatures, which certainly will help understand the origin of topological Hall effect, since it is not surprising for realizing neel-type Skyrmions based on CrTe₂. I don't have and bias on the last part of this work, since the results are clear enough, demonstrating a current induce magnetization change via the interface, which is of importance for future application in spintronics.

Reviewer #2:

Remarks to the Author:

In this paper, the authors performed a systematic characterization of epitaxial CrTe₂ films and CrTe₂/ZrTe₂ heterostructure. The work convincingly demonstrated the following noteworthy results:

1. confirming the high Curie temperature in monolayer CrTe₂ film through AMF measurements. The result was also proved by a different method in Ref. [32].
 2. Showing the irregular AHE behavior in CrTe₂ films as well as in CrTe₂/ZrTe₂ hybrids.
 3. demonstrating spin-orbit torque (SOT) in both CrTe₂/ZrTe₂ and Permalloy/ZrTe₂ hybrids.
- The results shed light on spintronics applications of CrTe₂-based 2D ferromagnets. The manuscript has been properly revised to address my previous questions. Therefore, I recommend the publication of this paper in Nature Communications.

Reviewer #3:

Remarks to the Author:

In this article, the authors report the study of an epitaxial van der Waals (vdW) heterostructure CrTe₂/ZrTe₂. Through MBE synthesis, vdW heterostructures with high-quality interfaces are prepared, confirmed by STEM and ARPES. In transport measurements, 1T-CrTe₂ grown on ZrTe₂ with one unit-cell thickness shows a clear anomalous Hall effect (AHE). In thicker layers, 12 unit-cell thick, AHE is observed at higher temperatures. However, the AHE becomes weaker and is replaced by an unconventional shape which is typical for the topological Hall effect (THE). The THE effect, however, is suppressed in thinner layers. The authors also demonstrate current-driven magnetization switching in device consisting of 3 unit-cell CrTe₂ and ZrTe₂.

In general, the authors present a careful study of the CrTe₂/ZrTe₂ epitaxial vdW heterostructures. Several interesting phenomena are observed. Although the experiments are well-conducted, the data clearly presented, and the paper well organized, I cannot recommend the publication of this article. First, CrTe₂ and ZrTe₂ are well-studied materials. Although the preparation of high-quality heterostructure with MBE has not been reported, the experimental results do not show any advantages in performance exceeding other spintronics systems. Second, from the study of the AHE and THE aspect, the authors do have some unconventional observations. However, they are not well studied and discussed.

In addition, I have some further questions that the authors need to consider.

1. About the THE observed in devices with 12 unit-cell CrTe₂, the authors posit that it is an inherent property of CrTe₂, considering the observations of THE in devices with interfaces with both sapphire and ZrTe₂. However, in a recent paper (Nat Commun 12, 809 (2021)), a sample with a similar thickness (10nm) does not show any sign of THE. In my opinion, the observed THE

might be strongly related to the interface. Although sapphire is not material with strong spin-orbit coupling, the interface between CrTe₂ and sapphire might still have strong interfacial Dzyaloshinskii-Moriya interaction (DMI). Similar observation has been reported in CrTe flake grown on STO (Nano Res. 11, 3116–3121 (2018)). I will suggest that the authors prepare CrTe₂ flakes with different thicknesses on sapphire and then compare their transport results.

2. Comparing CrTe₂ (12 uc) on ZrTe₂ to CrTe₂ on sapphire, one can find that CrTe₂/ZrTe₂ has stronger AHE but weaker THE. It seems that the coupling to ZrTe₂ facilitates AHE but hamper THE. This is quite different from CrTe₂/Bi₂Te₃. Can the authors discuss this difference, like preparing CrTe₂ with different thicknesses on ZrTe₂?

3. In the current-induced magnetization switching experiment, the authors claim that it is difficult to estimate SOT from current density in CrTe₂/ZrTe₂ device, due to the complicated nature of the domain nucleation and domain wall motion. However, this should be quite common in micron-size devices. Instead, the authors try to estimate the SOT in a ZrTe₂/permalloy device, where they get the SOT efficiency of about 0.014. This efficiency is actually quite small. The authors then claim that due to the non-ideal interface between ZrTe₂/Py, the value is a lower bound. Although these arguments are reasonable, no advantage is found in CrTe₂/ZrTe₂ heterostructure compared to other reported materials, which weakens the scientific interest of this study.

Response to Reviewers

“ZrTe₂/CrTe₂: an epitaxial van der Waals platform for spintronics”

Yongxi Ou *et al.*

Response to Reviewer #1’s specific comments/questions:

“The authors demonstrate an interesting epitaxy of heterojunction consisting of topological and 2d ferromagnetic materials. The the growth quality is high and the details are sufficient for repeating the results. The author also performed systematic hall measurement on the samples of pure CrTe₂ and CrTe₂/ZrTe₂ interfaces, revealing the anomalous Hall effect together with topological Hall effects. Before accept the author's explanation, I want the author display the magnetization results together with longitude magnetic resistance measurements for the above samples at corresponding temperatures, which certainly will help understand the origin of topological Hall effect, since it is not surprising for realizing neel-type Skyrmions based on CrTe₂. I don't have and bias on the last part of this work, since the results are clear enough, demonstrating a current induce magnetization change via the interface, which is of importance for future application in spintronics.”

We thank Reviewer #1 for her/his accurate summary of our results and the positive opinions. As suggested by the Reviewer, we have now added a new section in the Supplementary Materials including the longitudinal magnetoresistance measurements for a pure CrTe₂ film as well as the ZrTe₂/CrTe₂ samples at corresponding temperatures where they show the anomalous Hall effect. We have also included an example of a vibrating-sample-magnetometer-SQUID magnetometry measurement on one sample, sapphire/CrTe₂(12u.c.), in the same section together with an estimate of its magnetization. Our hesitation in presenting more detailed magnetometry measurements stems from the difficulty in reliably obtaining the magnetization of ultrathin CrTe₂ films after subtracting the significant diamagnetic substrate background signal in our measurements. At this stage, we believe the anomalous Hall data presented in the main text is a more reliable measure of the ferromagnetic order in our samples. We have emphasized our concerns about the reliability of the magnetometry measurements in the added section.

Response to Reviewer #2's specific comments/questions:

“In this paper, the authors performed a systematic characterization of epitaxial CrTe₂ films and CrTe₂/ZrTe₂ heterostructure. The work convincingly demonstrated the following noteworthy results:

- 1. confirming the high Curie temperature in monolayer CrTe₂ film through AMF measurements. The result was also proved by a different method in Ref. [32].*
- 2. Showing the irregular AHE behavior in CrTe₂ films as well as in CrTe₂/ZrTe₂ hybrids.*
- 3. demonstrating spin-orbit torque (SOT) in both CrTe₂/ZrTe₂ and Permalloy/ZrTe₂ hybrids.*

The results shed light on spintronics applications of CrTe₂-based 2D ferromagnets. The manuscript has been properly revised to address my previous questions. Therefore, I recommend the publication of this paper in Nature Communications.”

We thank Reviewer #2 for their careful reading of our paper and the accurate summary of our results. We are grateful for the Reviewer's recommendation that paper should be published in *Nature Communications*.

Response to Reviewer #3's specific comments/questions:

“In this article, the authors report the study of an epitaxial van der Waals (vdW) heterostructure CrTe₂/ZrTe₂. Through MBE synthesis, vdW heterostructures with high-quality interfaces are prepared, confirmed by STEM and ARPES. In transport measurements, 1T-CrTe₂ grown on ZrTe₂ with one unit-cell thickness shows a clear anomalous Hall effect (AHE). In thicker layers, 12 unit-cell thick, AHE is observed at higher temperatures. However, the AHE becomes weaker and is replaced by an unconventional shape which is typical for the topological Hall effect (THE). The THE effect, however, is suppressed in thinner layers. The authors also demonstrate current-driven magnetization switching in device consisting of 3 unit-cell CrTe₂ and ZrTe₂.

In general, the authors present a careful study of the CrTe₂/ZrTe₂ epitaxial vdW heterostructures. Several interesting phenomena are observed. Although the experiments are well-conducted, the data clearly presented, and the paper well organized, I cannot recommend the publication of this article. First, CrTe₂ and ZrTe₂ are well-studied materials. Although the preparation of high-quality heterostructure with MBE has not been reported, the experimental results do not show any advantages in performance exceeding other spintronics systems. Second, from the study of the AHE and THE aspect, the authors do have some unconventional observations. However, they are not well studied and discussed.”

We thank Reviewer #3 for a careful reading of our paper and the thoughtful comments. We are pleased to see that the Reviewer finds that our paper is “well-organized” and clearly presents data from “well-conducted” experiments, resulting in the observation of “several interesting phenomena.” While the Reviewer agrees that our work presents a careful study on the epitaxial ZrTe₂/CrTe₂ full vdW heterostructure, in their opinion, the two materials, ZrTe₂ and CrTe₂, have been separately “well-studied.” The Reviewer feels that the heterostructure does not show any “advantages in performance exceeding other spintronics systems” and that the “unconventional observations” in our paper were not well-studied and discussed. We respectfully disagree with this viewpoint.

We first address the Reviewer's comment that CrTe₂ is a “well-studied” 2D ferromagnet. We agree with the Reviewer that this material has attracted increased interest recently. However,

almost all the published studies on this material (cited in our manuscript) have focused on CrTe₂ flakes exfoliated from bulk crystals and some studies of CVD grown samples. To make some progress toward exploring the technological viability of this material, whether in the quasi-2D or purely 2D regime, studies will need to move beyond the exfoliation of small area samples and require thin film growth of complex multilayer stacks (heterostructures). As we mention in our introduction, there are only two reports of wafer-scale MBE growth of CrTe₂ thin films with good crystalline quality. Additionally, the basic understanding of the physics of CrTe₂ in the 2D and quasi-2D regime is still at a nascent stage and cannot be categorized as “well-studied.” Many rich physical properties of CrTe₂ (for example, chiral magnetic domains) remain to be fully explored and their full understanding requires access to well-controlled thin film growth. Our paper shows how to do this with heterostructures that allow magneto-transport measurements and will also elicit interest from experimentalists who want to probe such films with other techniques (such as magneto-optical spectroscopy, magneto-photogalvanic measurements, etc.) Our work also seeks to reveal hints of novel phenomena such as the THE, echoing the approach used in a contemporaneous independent study on a differently interfaced heterostructure, CrTe₂/Bi₂Te₃, cited in our manuscript (ACS Nano 15, 15710–15719 (2021)). Importantly, our work further demonstrates the robustness of the ferromagnetic order of CrTe₂ in the true 2D one-unit-cell limit when interfaced with another topological material (ZrTe₂). This paves the way for synthesizing heterostructures using CrTe₂ for potential 2D ferromagnet applications in topological spintronics.

We now address the Reviewer’s comment that ZrTe₂ is also “well-studied.” Although this is a candidate topological material (Dirac semimetal), there are very few published reports on its MBE growth and physical properties so far (all the relevant work is cited in our paper). In our work, we present the first comprehensive set of experimental and theoretical studies of the spintronic potential of ZrTe₂. This includes the first measurements of spin-charge conversion (via spin-torque FMR and current induced spin-orbit torque) and the first DFT calculation of the spin Hall conductivity. These results will be of great value for the spintronics community since studies of spin-charge conversion in topological semimetals (both Dirac and Weyl semimetals) are still at an early stage. In the early stages of a field such as this, obtaining the largest device efficiencies is not the highest priority. Rather, our study on the ZrTe₂/CrTe₂ heterostructure is aimed at introducing to the community the fundamental behavior of a thoroughly characterized, wafer-scale, well-controlled 2D ferromagnet/topological materials platform with clean interfaces to test

the spintronics concepts. We believe that the interesting results shown here will motivate other groups to pursue this materials platform.

The Reviewer also states that the unusual aspects of the AHE and THE observations in our work are not well studied and discussed. We provide a more elaborate response to this comment below in the context of the Reviewer's additional questions.

1) *“About the THE observed in devices with 12 unit-cell CrTe₂, the authors posit that it is an inherent property of CrTe₂, considering the observations of THE in devices with interfaces with both sapphire and ZrTe₂. However, in a recent paper (Nat Commun 12, 809 (2021)), a sample with a similar thickness (10nm) does not show any sign of THE. In my opinion, the observed THE might be strongly related to the interface. Although sapphire is not material with strong spin-orbit coupling, the interface between CrTe₂ and sapphire might still have strong interfacial Dzyaloshinskii-Moriya interaction (DMI). Similar observation has been reported in CrTe flake grown on STO (Nano Res. 11, 3116–3121 (2018)). I will suggest that the authors prepare CrTe₂ flakes with different thicknesses on sapphire and then compare their transport results.”*

We thank the Reviewer for raising an interesting and possibly viable explanation for the origin of the THE in our CrTe₂ thin films. In our manuscript, we had tentatively attributed the observed thickness dependence of the AHE and THE in our samples to the Coulomb screening effect in the 2D limit, referring to the study of CVD-grown CrTe₂ thin films referenced by the Reviewer (Nat Commun 12, 809 (2021)). As pointed out by the Reviewer, the CVD-grown films in that paper do not show a THE, perhaps because they are not interfaced with a substrate and thus are not subject to a Dzyaloshinskii-Moriya (DM) interaction. Thus, it is indeed possible that our observations indicate that we have an interfacially-induced DM interaction even when the substrate is sapphire. The Reviewer has pointed out that a thickness-dependent THE has been observed in another CrTe compound grown on SrTiO₃, another substrate that does not have an obvious spin-orbit coupling. We have cited the publication (Nano Res. 11, 3116–3121 (2018)) mentioned by the Reviewer in our revised manuscript. The Reviewer also suggests that we carry out a systematic study of the thickness dependence and substrate dependence of the THE in CrTe₂. However, since the focus of

our work here is on $\text{ZrTe}_2/\text{CrTe}_2$ heterostructures, such a study is beyond the scope of our current manuscript. It is indeed a very pertinent and interesting direction that the Reviewer suggests: we will be pursuing this in future work. In response to the Reviewer's suggestion, we have added the following statement in the manuscript.

“Our observation of a THE-like signature in CrTe_2 films grown directly on sapphire (a material that does not have strong spin-orbit coupling) suggests that the underlying physics is probably more complex. We might speculate that the interface between CrTe_2 and sapphire induces a DM interaction but we do not currently have any microscopic model to justify this speculation.”

2) *“Comparing CrTe_2 (12 uc) on ZrTe_2 to CrTe_2 on sapphire, one can find that $\text{CrTe}_2/\text{ZrTe}_2$ has stronger AHE but weaker THE. It seems that the coupling to ZrTe_2 facilitates AHE but hamper THE. This is quite different from $\text{CrTe}_2/\text{Bi}_2\text{Te}_3$. Can the authors discuss this difference, like preparing CrTe_2 with different thicknesses on ZrTe_2 ?”*

We thank the referee for again making another insightful observation. There is indeed a marked difference in the AHE and the THE in sapphire/ CrTe_2 (12u.c.) and $\text{ZrTe}_2/\text{CrTe}_2$ (12u.c.). In principle, this could be potentially due to the different interfacial contributions when CrTe_2 is interfaced with sapphire or with ZrTe_2 . Such an interfacial dependent AHE behavior is well known in other perpendicularly magnetized ferromagnetic heterostructures. On the other hand, we must be careful in jumping to conclusions without first examining simpler alternatives. Since ZrTe_2 is a semimetal with a conductivity similar to that of CrTe_2 , electrical transport measurements of $\text{ZrTe}_2/\text{CrTe}_2$ heterostructures necessarily involve current shunting through the non-magnetic ZrTe_2 layer. This can clearly affect the magnitude of the measured Hall effect and thus complicates direct comparisons between the amplitude of the THE signal in sapphire/ CrTe_2 (12u.c.) and $\text{ZrTe}_2/\text{CrTe}_2$ (12u.c.). We have added a sentence and a relevant reference in the revised manuscript when discussing the AH conductivity, pointing the readers to the possibility of interfacial contributions in the observed AHE and THE as well as cautioning about the effect of current shunting.

3) *“In the current-induced magnetization switching experiment, the authors claim that it is difficult to estimate SOT from current density in CrTe₂/ZrTe₂ device, due to the complicated nature of the domain nucleation and domain wall motion. However, this should be quite common in micron-size devices. Instead, the authors try to estimate the SOT in a ZrTe₂/permalloy device, where they get the SOT efficiency of about 0.014. This efficiency is actually quite small. The authors then claim that due to the non-ideal interface between ZrTe₂/Py, the value is a lower bound. Although these arguments are reasonable, no advantage is found in CrTe₂/ZrTe₂ heterostructure compared to other reported materials, which weakens the scientific interest of this study.”*

The Reviewer is correct that the literature has many reports of experiments that use switching current density to estimate the strength of SOT in some heterostructures. But this approach has its limitations and we are being cautious. As we mention in the manuscript, such estimates rely on the assumption of a macro-spin (single-domain) switching process. This assumption is likely violated in the 5 $\mu\text{m} \times 10 \mu\text{m}$ Hall bar devices used in our studies. Thus, any interpretation of our current induced magnetization switching data for SOT estimates would be compromised by complicated domain nucleation and domain motion. We also note that another commonly used SOT measurement on micron-size devices, the second harmonic response technique, is not suitable for our case, given that technique usually requires a large input current to generate the SOT. In our ultrathin CrTe₂ devices, this would result in artifacts due to both heating and thermal effects. Therefore, we performed ST-FMR measurements to estimate the strength of SOT in ZrTe₂, a result also supported by our DFT calculation.

The Reviewer notes that the estimated SOT efficiency (0.014) in ZrTe₂/Py is too small to be competitive for technological interest. However, as we noted in the manuscript, this is only a lower bound because of the less-than-ideal ZrTe₂/Py interface and possibly the location of the chemical potential (Fermi energy). The DFT calculation in our paper shows that the strength and the sign of the spin Hall conductivity in ZrTe₂ is a strong function of the chemical potential. Thus, we anticipate that we will be able to increase the SOT efficiency by growing heterostructures wherein the chemical potential is optimized. In the end, the technological potential of ZrTe₂ for spintronics applications (such as MRAM) will depend on the threshold current density and power dissipated during magnetization switching. Our measurements already show that the threshold switching

current density in $\text{ZrTe}_2/\text{CrTe}_2$ can be comparable to that in other heavy-metal-based systems. This demonstrates that the strength of SOT from ZrTe_2 is significant enough to manipulate the magnetic states of CrTe_2 , a key feature for SOT spintronics. Thus, the $\text{ZrTe}_2/\text{CrTe}_2$ heterostructure we have developed provides a valuable, well-controlled material platform with a clean interface for fundamental scientific studies that explore the role of 2D ferromagnets and topological materials for topological spintronics applications. Our intent is to highlight the potential of this heterostructure to stimulate further studies of this material in the spintronic community.

Reviewers' Comments:

Reviewer #1:

Remarks to the Author:

The author showed the transport curves and magnetization loops at low temperatures, displaying consistent features as the hall measurements. Now I was convinced by the author for the claims of this article. I think the article is ready for publication in my opinion.

Reviewer #3:

Remarks to the Author:

In the response letter, the authors provide a detailed statement on the novelty of their study and point-to-point answers to my questions. Meanwhile, the manuscript is revised accordingly. I think the response well clarifies all my concerns and questions. Therefore, I would like to recommend the publication of this work in Nature Communications.

Response to Reviewer #1

Reviewer #1 (Remarks to the Author):

The author showed the transport curves and magnetization loops at low temperatures, displaying consistent features as the hall measurements. Now I was convinced by the author for the claims of this article. I think the article is ready for publication in my opinion.

We are delighted to see that reviewer thinks the paper is ready for publication. We thank the referee for their useful review.

.....

Response to Reviewer #3:

Reviewer #3 (Remarks to the Author):

In the response letter, the authors provide a detailed statement on the novelty of their study and point-to-point answers to my questions. Meanwhile, the manuscript is revised accordingly. I think the response well clarifies all my concerns and questions. Therefore, I would like to recommend the publication of this work in Nature Communications.

We are delighted to see that reviewer thinks the paper is ready for publication. We thank the referee for their useful review.